# Evaluation of Recovery Methods for *Fragaria vesca* L. Oil: Characteristics, Stability and Bioactive Potential

**DOI:** 10.3390/foods12091852

**Published:** 2023-04-29

**Authors:** Magdalena Grajzer, Benita Wiatrak, Paulina Jawień, Łukasz Marczak, Anna Wojakowska, Rafał Wiejak, Edward Rój, Wojciech Grzebieluch, Anna Prescha

**Affiliations:** 1Department of Dietetics and Bromatology, Wroclaw Medical University, Borowska 211, 50-556 Wroclaw, Poland; 2Department of Pharmacology, Wroclaw Medical University, Mikulicza-Radeckiego 2, 50-345 Wroclaw, Poland; benita.wiatrak@umw.edu.pl; 3Department of Biostructure and Animal Physiology, Wroclaw University of Environmental and Life Sciences, Norwida 25/27, 50-375 Wroclaw, Poland; paulina.jawien@upwr.edu.pl; 4European Centre for Bioinformatics and Genomics, Institute of Bioorganic Chemistry, Polish Academy of Sciences, 61-138 Poznanń, Poland; lukasmar@ibch.poznan.pl (Ł.M.); astasz@ibch.poznan.pl (A.W.); 5Research Group Supercritical Extraction, Łukasiewicz Research Network–New Chemical Syntheses Institute, Al. Tysiąclecia Państwa Polskiego 13a, 24-110 Puławy, Poland; 6Department of Conservative Dentistry with Endodontics, Wroclaw Medical University, Krakowska 26, 50-425 Wroclaw, Poland; wojciech.grzebieluch@umw.edu.pl

**Keywords:** *Fragaria vesca*, seed oil, cold pressing, supercritical CO_2_ extraction, antioxidants, oxidative stability, cytotoxicity studies

## Abstract

Wild strawberry *(Fragaria vesca* L.) seed oil (WSO) recovered by two methods—cold pressing (CP) and extraction with supercritical carbon dioxide (SCO_2_E)—taking into account the different extraction times, was characterized for its composition and quality. The cytotoxicity assessment of WSOs was also carried out using the normal human dermal fibroblast (NHDF) cell line. Tocopherol and total polyphenol contents were significantly higher in WSO recovered by SCO_2_E, up to 1901.0 and 58.5 mg/kg, respectively, in comparison with CP oil. In CP oil, the highest content of carotenoids and squalene was determined (123.8 and 31.4 mg/kg, respectively). Phytosterol summed up to 5396 mg/kg in WSO collected in 30 min of SCO_2_E. Moreover, the highest oxidative stability was found for this oil. All studied WSOs were non-cytotoxic in lactate dehydrogenase (LDH) leaching and sulforhodamine B (SRB) assays; however, oils collected by SCO_2_E in 15 and 30 min were found to be cytotoxic in the tetrazolium salt (MTT) test, with the CC_50_ at a concentration of 3.4 and 5.5%, respectively. In conclusion, the composition of WSO indicates that, depending on the method of its recovery, seeds can have different bio-potencies and various applications.

## 1. Introduction

Wild strawberry *(Fragaria vesca* L.) is a species from the *Rosaceae* family, widely distributed throughout Europe, Northern America and Northern Asia [1]. Fruits of *F. vesca* are renowned for their exquisite taste and for the high antioxidant activity linked to the content of phenolics, especially ellagitannins and flavonols [2,3]. The fruits are usually consumed fresh or processed into juice, syrup and jam. Washing and processing fruits generates leftovers liquid containing a range of biomass that can be recovered for biomedical applications. Seeds as part of the residual biomass from processing of wild strawberry fruits contain about 20% fat rich in polyunsaturated fatty acids (PUFA, up to 80% of total fatty acids) with a low ratio of n-6 to n-3 fatty acids (less than 1.5). This renders wild strawberry oil (WSO) an interesting specialty oil; however, to fully appraise its value, analysis of minor oil components coextracted with acylglycerols is required. Yet, literature-derived data on the composition of WSO do not include information on the content of bioactive oil’s micronutrients, such as phytosterols, tocochromanols, carotenoids and phenolic compounds [4]. Our group and other researchers have shown that the micronutrient content of oils recovered from *Rosaceae* seeds largely determines their bioactivity in both in vitro [5,6] and in vivo systems [7,8].

The method of oil extraction from the raw material has a significant impact on the composition of vegetable oils. While major oils are mostly recovered using methods of high effectiveness, such as expeller pressing involving heating of the material and additional solvent extraction followed by multi-stage refining, minor and specialty oils are produced using environmentally friendly alternatives: cold pressing and supercritical extraction [9,10]. Cold pressing (CP) is a commonly used method for manufacturing oils with high nutritive and functional properties. The advantage of CP is the use of a simple and almost waste-free technology, allowing substantial preservation of the non-acylglycerol fraction of the oil. The disadvantage of using this method is the exposure of oil to oxidation during the process and its low efficiency, resulting in 5–15% oil remaining in the pomace. Extraction of oils with supercritical carbon dioxide (SCO_2_E) can be an attractive alternative to the production of vegetable oils that is carried out in an anaerobic atmosphere with almost 100% recovery of oil from seeds [2]. The oil obtained in this way is distinguished by the particularly rich composition of its non-acylglycerol fraction, abundant in antioxidant and health-promoting compounds.

Therefore, this study aimed to compare the phytochemical characteristics and oxidative stability of WSO recovered by two methods: CP and SCO_2_E. Due to the fact that the SCO_2_E allows convenient fractionation of the yield with varying composition, the extracts were collected at consecutive time points in the run. As WSOs have not yet been considered for food or biomedical use, cell toxicity tests were carried out, including three methods assessing WSO impact on cell membrane integrity, cell mitochondrial metabolic activity and overall cell viability/proliferative ability using human dermal fibroblast cells (NHDF cell line).

## 2. Materials and Methods

### 2.1. Samples

All oil samples were recovered from the same batch of seeds purchased from the Polish manufacturer OleoWita Ewelina Narożna-Domska, either by cold pressing (CP) or supercritical carbon dioxide extraction (SCO_2_E). Cold pressing was performed in a hydraulic press under nitrogen pressure by OleoWita Ewelina Narożna-Domska (Milicz, Poland) (Figure 1), while SCO_2_E was performed by the Research Group of Supercritical Extraction at Łukasiewicz Research Network—New Chemical Syntheses Institute (Puławy, Poland) (Figure 2).

Due to the planned cooperation with industry in the production of wild strawberry seed extract, cost optimization was carried out, which showed that the optimal parameters were as described in Table 1.

During extraction with supercritical carbon dioxide, WSO was collected at different time intervals (first every 15 min: at 15, 30, 45, 60, 75 min, then every 30 min: at 105 and 135 min). As a result, 7 fractions were obtained, which were denoted SCO_2_15, SCO_2_30, SCO_2_45, SCO_2_60, SCO_2_75, SCO_2_105 and SCO_2_135, respectively. This process was repeated three times. The practical criterion for the completion of the process was a cumulative increase in the mass of the extract below 0.3%. The yield increment during the SCO_2_E is presented in Figure 3. The extracts collected were subjected to quality analysis, including fatty acid composition, antiradical capacity and quality parameters. Fractions that had the best quality parameters were subjected to minor compounds analysis and accelerated oxidative stability tests. For this purpose, individual samples at each time point were carefully mixed and prepared for analysis in accordance with the methodology used. Oil samples were stored at −20 °C until analysis.

### 2.2. Oils’ Characteristics

#### 2.2.1. Fatty Acid Composition

Fatty acids of oils were converted into methyl esters derivatives according to Prescha et al. [11]. The chromatographic separation was performed on a 6890N gas chromatograph (Agilent Technologies, Santa Clara, CA, USA) equipped with an FID detector and a Supelco SPTM-2560 fused silica capillary column, 100 m × 0.25 mm × 0.2 μm (Bellefonte, PA, USA), as described previously by [5].

#### 2.2.2. Antiradical Capacity

Measurements of the antiradical activity of the tested oils were carried out using the 2,2-diphenyl-1-picrylhydrazyl radical (DPPH). Results were obtained spectrophotometrically according to the method described by Tuberoso et al. [12]. The DPPH test was performed for both the oil samples and their hydrophilic (HF) and lipophilic (LF) fractions. Individual fractions were separated by the method developed by Espín et al. [13]. The results were expressed as Trolox equivalent antiradical capacity (TEAC).

#### 2.2.3. Quality Parameters

Acid value (AV) and peroxide value (PV) were determined according to the official Polish methods [6].

#### 2.2.4. Phytosterol and Squalene Contents

The oils were saponified and the unsaponified fraction was silylated. The silyl-derivatives of phytosterols as well as squalene were determined on a 7890A gas chromatograph equipped with an electron ionization (EI) source and the autosampler system PAL3 RSI (CTC Analytics AG, Zwingen, Switzerland) coupled to a 5975C mass spectrometer (Agilent Technology, USA). The components were separated on a DB-5MS capillary column, 30 m × 0.25 mm × 0.1 μm (J & W Scientific, Folsom, CA, USA) [5]. Helium at a flow rate of 1.1 mL/min was used as the carrier gas. The source temperature was 230 °C and the temperature of the transfer line was 290 °C; the separation started at 120 °C, was held for 1 min, then the temperature was increased at the rate of 5 °C/min to the final temperature of 290 °C, held for 15 min. Identification of individual phytosterols and squalene was carried out using mass spectra and retention times of external standards. Oil samples were prepared for measurements according to the modified method of Shukla et al. [14].

#### 2.2.5. Tocopherol and Carotenoid Contents

In order to analyze tocopherol and carotenoid profile, oil samples were saponified and the unsaponifiable fraction was collected, then the lipophilic compounds were extracted according to a protocol described previously by Fromm et al. [15]. Compounds were separated on a carotenoid column (YMC, 3.0 µm 150 mm × 3.0 mm, BCS Bujno Chemicals, Warsaw, Poland) connected to HPLC Agilent Technologies 1260 Infinity II. Elution was carried out with the following solvents: line A—acetonitrile: methanol: water (15:81:4 *v*/*v*); line B—acetonitrile: methanol (50:50 *v*/*v*); flow rate of the mobile phase—1 mL/min; gradient from line A for 45 min to 100% of line B. Identification of tocopherols and carotenoids was made by comparing their retention times and spectra with the retention times and spectra of the respective standards and based on external standard curves in the concentration range of 0.01–0.2 mg/mL individual standards.

#### 2.2.6. Polyphenolic Extraction and Analysis

Isolation of polyphenols from oil was accomplished using the method developed by Pirisi et al. [16].

Samples were analyzed using an LC/MS system based on a Dionex RSLC 3000 UPLC system (Thermo Scientific, Bremen, Germany) connected with a timsTOF Pro mass spectrometer (Bruker Daltonics, Bremen, Germany). Analytes were separated on a Zorbax Eclipse XDB-C18 column (2.1 × 150 mm, 3.5 µm). Chromatographic separation on the UPLC system was performed at a flow rate of 0.5 mL/min using mixtures of the following two solvents: A (99.9% H_2_O, 0.1% formic acid (*v*/*v*)); B (99.9% acetonitrile, 0.1% formic acid (*v*/*v*)). The elution steps were as follows: 0–2 min isocratic separation at 5% B; 3–21 min linear gradient to 95% B; 22–23 min of isocratic flow at 95% B and a return to the initial conditions for an additional 3 min.

The timsTOF Pro spectrometer consisted of ESI operating at −4.5 kV, nebulization with nitrogen at 1.6 bar and a dry gas flow of 8.0 L/min at a temperature of 220 °C. The system was calibrated externally using a calibration mixture containing sodium formate clusters. Additional internal calibrations were performed for every run by injecting the calibration mixture using a divert valve during LC separation. All calculations were performed using the HPC quadratic algorithm. These calibrations yielded an accuracy of less than 5 ppm. MS/MS spectra were acquired as data dependent scans with the maximum frequency of 5 MS/MS spectra per one MS survey scan. The collision energy was dependent on the molecular masses of the compounds and was set between 10 and 25 eV. The instrument operated at a resolution higher than 30,000 (FWHM, full width at half maximum) using the program otofControl version 6.2, and data were analyzed using the DataAnalysis version 6.0 package, which was supplied by Bruker. All profiles were registered in both positive and negative ion modes.

Raw data files (.d) were converted to abf format for analysis using MSDial software (v. 4.92). For compound identification using MSDial, the MSP database from the CompMS site containing 324,191 records in positive mode and 44,669 records in negative ion mode was used. Identified artifacts were excluded from further analyses and the obtained normalized (using the total ion current (TIC) approach) results were exported to Excel for pre-formatting and then used for statistical analyses. Statistical analyses and hierarchical clustering of data (HCA) were performed using Perseus software ver. 1.6.15.0. For HCA, data matrix was reduced to contain only compounds of interest, values were transformed using Z-score algorithm to scale the data correctly. A color gradient was used to represent normalized values, where blue color indicated a lower amount and red color corresponded to a higher amount of compound.

Total polyphenols were measured calorimetrically using the Folin–Ciocalteu reagent, with caffeic acid as a standard [12].

#### 2.2.7. Oxidative Stability Studies

For the study of the oxidative stability of oils, a Perkin Elmer differential scanning calorimeter (Pyris 6) was used. The standard procedure of calibration of the temperature scale was performed using In and Zn standards. The oil sample (3–5 mg) was placed in an open pan filled with alumina (0.015 g) and after the cell was equilibrated at 30 °C, the temperature program was set. The sample was purged in oxygen at flow of 50 mL/min and nitrogen at flow of 20 mL/min. In the isothermal procedure, the measurements were performed at a constant temperature of 100, 110, or 120 °C. In the dynamic (non-isothermal) procedure, the temperature ranged from 50 to 350 °C at a rate of 5 °C/min, 10 °C/min or 15 °C/min. Exothermic peaks in the DSC thermograms were oriented downwards. The DSC kinetic parameters were determined from each scan by the Pyris 6 calculation software package.

### 2.3. Cytotoxicity Assays

#### 2.3.1. Cell Culture

Normal human dermal fibroblast cell line, NHDF (CC-2511), was purchased from Lonza (Basel, Switzerland). Cells were grown in culture flasks (T-180, TPP, Sigma-Aldrich, Saint Louis, MO, USA) in Dulbecco’s Modified Eagle Medium without phenol red (DMEM; no phenol red; Gibco, Thermo Fisher Scientific, Waltham, MA, USA) supplemented with 2 mM L-glutamine (Gibco, Thermo Fisher Scientific, Waltham, MA, USA). The cell culture medium was supplemented with 10% fetal bovine serum (FBS; Gibco, Thermo Fisher Scientific, Waltham, MA, USA) and 1% antibiotic solution containing 10,000 penicillin units/mL, 10 mg/mL streptomycin and 25 µg/mL amphotericin B (Sigma-Aldrich, Saint Louis, MO, USA). Every three days, the cell culture was assessed under a light microscope (EVOS FL, Thermo Fisher Scientific, Waltham, MA, USA), and if the confluence was less than 80%, the medium was renewed; otherwise, the cell culture was passaged using TrypLE solution (Gibco, Thermo Fisher Scientific, Waltham, MA, USA). Cells were grown under standard culture conditions at 37 °C in an incubator in 95% humidified air containing 5% CO_2_. To reduce the number of cells or to seed on multi-well test plates, the culture medium from the culture flask was removed, the culture medium was washed with TrypLE to inactivate FBS, and then fresh TrypLE was added for 2–3 min at 37 °C to detach the cells from the surface of the culture flasks. After this time, fresh medium was added to inactivate the TrypLE. The suspension was collected and centrifuged to remove the TrypLE solution. After suspending the cell culture in fresh medium, the suspended cells were collected. A 25 µL sample of the cell suspension was stained with the same volume of 0.4% trypan blue solution and counted in a Brucker chamber under an EVOS FL microscope.

A 100 µL sample of medium with suspended cells at a density of 1.0 × 10^5^ was placed in each well of a 96-well plate (TPP, Sigma-Aldrich, Saint Louis, MO, USA). After plating, cells were left for 24 h in a CO_2_ incubator for cell attachment and homeostasis. The cell culture medium was removed from the wells and replaced with 100 µL of fresh cell culture medium supplemented with WSOs. The concentration of the WSOs was 0.5, 1.0, 2.5, or 5.0% for the next 24 h. This experiment was performed in three series using cells from different cell passages. Each series consisted of 5 replicates corresponding to different growth conditions (variable concentration and type of WSO extract).

#### 2.3.2. Viability Measurements with the MTT Method

In the MTT assay, according to ISO 10993 part 5, cytotoxicity was assessed with tetrazolium salt (3-[4,5-dimethylthiazol-2-yl]-2,5-diphenyltetrazolium bromide, MTT, Sigma-Aldrich, Saint Louis, MO, USA). The yellow MTT is reduced to purple crystals by succinate dehydrogenase, which were then dissolved in isopropanol. Additionally, the tetrazolium salt is reduced by NADH or NADPH in microsomes and cytosol. Cell morphology was also assessed subjectively according to a 4-point scale using an inverted microscope [17,18].

The observations of cells after 24 h incubation with WSOs was performed using a microscope (EVOS FL). The culture medium was removed from the wells, and 50 μL of 1.0 mg/mL MTT solution in PBS buffer was added. After 2 h incubation at 37 °C, isopropanol was added to dissolve formazan crystals formed in cells as a result of redox potential. Absorbance was measured at 570 nm using the VarioSkan Go reader (Thermo Scientific, Waltham, MA, USA).

The cell viability was calculated according to the following formula: V (%) = OD570A/OD570B, where V is the cell viability [%], OD570A is the average value of the measured optical density of the tested sample and OD570B is the average value of the measured optical density of the blank (culture only in a medium without examinations).

#### 2.3.3. Cytotoxicity Measurements with the LDH Leakage Assay

Cytotoxicity evaluation of lactate dehydrogenase (LDH) leaching into the culture medium was performed from the pooled supernatant from the MTT assay. The collected medium was centrifuged at 3000× *g* rpm for 5 min to obtain a cell-free supernatant. Then, 30 µL of the centrifuged supernatant was placed in 96-well plates, to which 100 µL of the reaction mixture (686 µM iodonitrotetrazolium chloride (INT), 291 µM phenazine methyl sulfate (PMS), 1.35 mM B-nicotinamide adenine dinucleotide (NAD+), 200 mM Tris (pH 8.2) and 55.5 mM lithium L-lactate) was added. Plates were left in the dark for 20 min at room temperature. At the end of the incubation, absorbance was measured using a VarioSkan Go microplate reader (Thermo Scientific, Waltham, MA, USA) at 490 nm and 680 nm.

#### 2.3.4. Viability/Proliferative Ability Measurements with the SRB Assay (Total Protein Content)

The total protein content test uses sulforhodamine B dye, which binds to cell protein at the appropriate pH. Cell cultures were incubated for 24 h in complete medium for 24 h in 96-well plates to adhere the cells to the surface of the wells. One plate was then fixed with a 50% TCA solution at 4 °C as a control to assess the amount of cell protein before adding the test oils. WSOs were added to the remaining plates for the next 48 h. After this time, the cultures were fixed with cold 50% TCA at 4 °C for 1 h. Then, all plates were washed 5 times under running water and dried. In the next step, the cells were stained at room temperature for 30 min with a 0.4% solution of sulforhodamine B in 1% acetic acid. The unbound dye was removed by washing five times with 1% acetic acid. After drying the plates, the dye attached to the proteins was dissolved in 10 mM Trizma base. Absorbance was measured at 490 nm using a VarioSkan Go reader (Thermo Scientific, Waltham, MA, USA).

### 2.4. Statistical Analysis

The multiple comparisons of data obtained in phytochemical and oxidative stability analyses were performed using one-way ANOVA and Tukey’s test. Differences between oils were considered statistically significant at *p* < 0.05. Based on the obtained biological results, the concentration that inhibits 50% of cell viability (CC_50_) was determined. For this purpose, a non-linear regression was determined using the dependence of the biological effect on the molar concentrations of compounds (four-parameter logistic model with Hill slope).

## 3. Results

### 3.1. Oils’ Chemical Characteristics

#### 3.1.1. Fatty Acid Composition

Table 2 shows the fatty acid composition of WSOs recovered by CP or SCO_2_E, with the latter collecting seven extracts at successive time points. All the WSO extracts obtained by SCO_2_E were characterized by a higher α-linolenic acid (C18:3 n-3) content (% of the fatty acids) and lower oleic (C18:1 n-9) and linoleic acid (C18:2 n-6) percentage than in CP oil. Among SCO_2_E oils, the most abundant in α-linolenic acid were SCO_2_30 and SCO_2_45 (ca. 43.3–43.4%), and for these extracts, the highest sum of PUFA and the lowest n-6/n-3 fatty acids ratio and the lowest oleic acid contents were found.

#### 3.1.2. DPPH Radical Scavenging Activity of WSOs

Table 2 shows the antioxidant activity expressed in TEAC, for oils not subjected to fractionation, along with their hydrophilic and lipophilic fractions. The highest antiradical capacity measured in not-fractionated oils was found for SCO_2_15 and CSO_2_30 (1.24 and 1.26 mm TEAC/kg, respectively) and the lowest for SCO_2_60 (0.77 mM TEAC/kg). When the lipophilic and hydrophilic fractions were tested, SCO_2_30 showed the highest antiradical activity among all tested oils (1.28 and 0.56 mm TEAC/kg, respectively).

#### 3.1.3. Quality Parameters

The acid (AV) and peroxide values (PV) of CP and SCO_2_E WSOs are shown in Table 2. The AV ranged from 0.68 in SCO_2_45 to 1.29 in SCO_2_15 and did not exceed the limit of free fatty acid content established by the Codex Alimentarius Commission (CAC) standard for cold-pressed and virgin oils (up to 4 mg KOH/g of oil). All studied WSOs were also characterized by PV values not exceeding 15 mEq O_2_/kg which is the CAC upper limit for cold-pressed oils and were the lowest in SCO_2_15 and SCO_2_45 (ca. 1.3 mEq O_2_/kg) and CP oil (1.68 mEq O_2_/kg). However, SCO_2_E oils collected at intervals over 45 min were characterized by significantly higher PV value than other WSOs studied (8.77–12.91 mEq O_2_/kg).

Due to the more favorable basic characteristics, namely, the composition of fatty acids and low values of the oxidation parameter (PV, AV), of the WSOs recovered by SCO_2_E, only SCO_2_15, SCO_2_30 and SCO_2_45 were taken for further analyses of the minor components and for the in vitro tests.

#### 3.1.4. Phytosterol and Squalene Content

Table 3 presents the phytosterol composition in WSO recovered by CP and in first three WSO samples collected during SCO_2_E. The main sterol component of WSO oils was β-sitosterol, which accounted for 94% of average of total phytosterol content. The WSO richest in phytosterols was SCO_2_30, followed by SCO_2_45. The amount of squalene was low, up to 31.4 mg/kg in WSO obtained by CP.

#### 3.1.5. Tocopherol and Carotenoid Contents

Tocopherol and carotenoid compositions of CP WSO and selected WSOs obtained by SCO_2_E are presented in Table 3. The total amount of tocopherols varied from 402 mg/kg in SCO_2_45 to 1901 mg/kg in SCO_2_15. The main tocopherol was γ-tocopherol, which in CP oil accounted for 38% and in SCO_2_30 up to 54% of the total tocopherol amount. In oils recovered by SCO_2_E, the second most abundant tocopherol isomer was δ-tocopherol, especially in SCO_2_15 and SCO_2_30. Moreover, β-tocopherol was only present in SCO_2_E oils.

The highest total carotenoid content was in CP oil (123.8 mg/kg). In SCO_2_E oils the number of carotenoids decreased with extraction time. The main carotenoid in CP oil and SCO_2_15 was all-*trans*-β-carotene, which accounted for 46% and 52%, respectively, followed by all-*trans*-zeaxanthin (32% and 38%). CP oil contained all-trans-α-cryptoxanthin, absent in the SCO_2_E oils. In SCO_2_30 and SCO_2_45, the major carotenoids were all-*trans*-zeaxanthin (51% and 59%, respectively) and the total amount of carotenoids was significantly lower than in CP oil and SCO_2_15.

#### 3.1.6. Polyphenol Content and Composition

The total polyphenol content in CP and selected WSOs is presented in Table 3. All the selected oils recovered by SCO_2_E contained significantly more of these compounds than CP oil, with SCO_2_15 being the richest in these compounds (58.45 mg/kg). Therefore, a thorough analysis of polyphenol composition was performed in SCO_2_15 and related to CP oil as a reference. The polyphenol profiling based on signal intensity in ESI-negative and -positive mode is presented in Figure 4, and the content of the individual compounds in the polyphenol group is presented in Appendix A. As the analysis of compounds occurring in negligible amounts allows a greater tolerance for standard deviation (up to 15% of average value), for a more reliable representation of the results obtained, the polyphenol profile in each of the four replicates performed for these oils is presented using hierarchical clustering of data. For SCO_2_15, remarkable differences were noted in polyphenol profile when compared to CP oil. The most abundant components in WSO produced by SCO_2_E were the derivatives of hydroxybenzoic and cinnamic acids as well as chrysin, apigenin and daphnetin, where the polyphenol fraction of WSO recovered by CP was dominated by liquiritin and, in most of the sample replicates (3), 4-hydroxybenzoate.

#### 3.1.7. Oxidative Stability Studies

Table 4 shows the results of accelerated oxidation tests performed on CP and selected SCO_2_E WSOs at constant and increasing temperature. From the thermograms recorded in isothermal conditions, the oxidative induction time (OIT) was derived, which indicates the start of lipid oxidation. SCO_2_30 was characterized by the highest OIT values of temperature 100 and 110 °C; the second most oxidation-resistant oil in isothermal conditions was oil recovered by CP. SCO_2_45 turned out to be extremely susceptible to oxidation at all temperatures used. The propagation phase, when lipid peroxy radicals are formed in a chain reaction and lipid hydroperoxides accumulate in oil, was the longest in CP, but only at 100 °C. At higher temperatures, propagation was slower in SCO_2_30. Under dynamic conditions, at each of the applied temperature rates the oxidation induction temperature (T_on_) and propagation temperature range obtained for the tested oils were less differentiated than under isothermal conditions. However, also under dynamic conditions, the lowest oxidative stability of SCO_2_45 was confirmed. Activation energy calculated for studied oils, which is the energy necessary to initiate induction of the oxidation process [19], calculated for the studied oils was similarly high in the case of CP oil and SCO_2_30, while a significantly lower value was obtained for SCO_2_15. OIT values noted for SCO_2_45 did not allow the Ea to be determined for this oil.

### 3.2. In Vitro Studies

The cytotoxic activity of selected WSOs was assessed in vitro against the normal cell line (NHDF) using the standard fluorometric assays, which determine various manifestations of cell life. The viability test uses the MTT method in accordance with the ISO 10993 part 5 standard. Lactate dehydrogenase leakage assay (LDH) is based on the release of lactate dehydrogenase outside the cells, indicating damage to the cell membrane. In the SRB assay, the total amount of cellular protein is measured. The estimated 50% maximum cytotoxic concentrations (CC_50_) for the tested oils are shown in Table 5. In the LDH and SRB tests, no cytotoxic effect was observed for any of the tested oils. Regarding the toxicity to the NHDF cell line in the MTT test, it was observed that WSO recovered by CP and by SCO_2_E for 45 min were non-toxic. Oils recovered by SCO_2_E for a shorter period of time showed some toxicity toward the NHDF cell line, with CC_50_ values of 3.35% and 5.49% at 15 and 30 min of extraction, respectively. Generally, the cytotoxicity of oils against normal human fibroblasts was inversely related to the time of its recovery by the SCO_2_ method.

The results of the morphological evaluation for the concentration of 5% of the tested oils are presented in Table 6. The morphological assessment was performed for 5% of the tested oils because, at lower concentrations, the cytotoxicity degree is 0 according to the ISO standard. The CP and SCO_2_45 oils did not induce morphological changes in the cultured cells. They did not reduce the cell confluence compared to the blank culture. However, in the case of oil recovered in a shorter SCO_2_E time, greater changes in morphology and decreased cell viability were observed. Cell morphological changes after contact with the tested oils are shown in Figure 5.

## 4. Discussion

The discussed research investigated the phytochemical characteristics and oxidative stability studies of WSO recovered by two methods: cold pressing (CP) and extraction by supercritical CO_2_ (SCO_2_E). It also assessed the potential toxicity of emulsions obtained from these oils in an in vitro model.

The CP method in oil production has been valued for years as environmentally friendly and low-cost. However, the SCO_2_E method, although associated with higher investment costs, is becoming an important alternative in oil production, not least because of its high oil yields and conditions that protect an oil from deterioration [18]. This method, through the use of appropriate extraction conditions, allows a product with the desired profile to be obtained, thereby broadening the range of applications of the raw material in the food, cosmetic and pharmaceutical industries [20,21]. The few and limited studies already carried out indicated that WSO could be a valuable raw material for many applications [4,22]. However, due to the expected rich composition of the WSO, in this study we decided not only to compare the effects on the constituent profile of the two alternative production methods, but also to investigate the effect of one of the SCO_2_E parameters, which is the yield collection time.

In terms of its phytochemical characteristics, WSO was an excellent source of essential fatty acids, especially α-linolenic acid, which is deficient in Western diets, but also other components of unsaponifiable oil fractions: tocopherols, carotenoids, polyphenols and sterols, regardless of the recovery method used in the work. However, the SCO_2_E influenced the fatty acid composition, and preserved total PUFA and particularly α-linolenic acid, increasing its content by up to 29% in comparison to CP oil. The high ALA content was not accompanied by a high PV value in CP oil and SCO_2_E oils extracted up to 45 min. In the case of CP, this is indicative of both the low amount of lipid hydroperoxides in the raw material and the minor influence of pressing conditions on oil oxidation. SCO_2_E is carried out under oxygen-free conditions, so the PV amounts are not attributable to the incidence of oxidation processes during extraction. However, the differences in PV between the SCO_2_E oils are due to the high selectivity of CO_2_ for lipophilic components [20]. As a result, the hydroperoxides, which have a higher polarity than parent fatty acids, were not extracted with the early oily extracts. In the late extracts, on the other hand, the ratio of lipophilic to less-lipophilic substances was lower, hence, the greater amount of oxidation products in the oils.

Generally, the fatty acid composition of WSO recovered by SCO_2_E was in accordance with that obtained by other authors in WSO by chloroform/methanol extractions [4,22]. Moreover, the n-6/n-3 ratio makes this oil a valuable source of essential fatty acids in the human diet.

The recovery method influenced the concentration of microcomponents in CP oil and three selected SCO_2_E products yielded between 15 and 45 min of extraction. SCO_2_E, regardless of the time of extraction, has been shown to better preserve not only PUFA, but also polyphenols in comparison with CP. Additionally, a significant, several times higher recovery of polyphenols using SCO_2_E in relation to CP was observed in our earlier study of the composition of raspberry seed oil [5]. This confirms the high penetrating capacity of CO_2_ in the release of polyphenols from the oily raw material [23]. It should be emphasized that our work is, to the best of our knowledge, the first to analyze the composition of polyphenols in WSOs recovered by both CP and SCO_2_E. CP oil not only had a significantly lower toral polyphenols content, in addition, its composition was dominated by only two compounds: 4-hydroxybenzoate and liquiritin (a flavone). In SCO_2_E oil, a wide range of phenolic compounds were determined. Among them were several derivatives of cinnamic and p-hydroxybenzoic acids that have been detected in many other seed oils of the Rosaceae family [5,6]. In addition, the SCO_2_E method enabled the extraction of seed flavonoid compounds, including chrysin, 7-methylchrysin, naringenin chalcone, apigenin and daphnetin, into the WSO. According to a recently published analysis of phenolics in berry seeds, including *F. ananassa* but not *F. vesca*, these raw oily materials may accumulate 26 different flavonoids in total, among them chrysin and apigenin [24].

In the present study, we used constant conditions of pressure, temperature and CO_2_ flow, which were found previously to be optimal for obtaining oil with the highest concentrations of phytosterols and tocopherols [20,25,26]. However, it seems that the duration of the SCO_2_E process could be of significance in recovery of these bioactive compounds, with the retrieval depending on the chemical nature of the compound. In the present study, it seems that the optimum duration of the extraction process was between 15 and 30 min. The oils collected in this period of time of SCO_2_E stood out in terms of tocopherols (SCO_2_15 and SCO_2_30), polyphenols (SCO_2_15) and phytosterols (SCO_2_30). In addition, a particularly high amount of δ-tocopherol was found in both SCO_2_15 and SCO_2_30, and these amounts were significantly higher than in the oil obtained by the CP method and the SCO_2_45. In contrast, these two SCO_2_E oils contained less α-tocopherol than CP and SCO_2_45 oils. Moreover, SCO_2_45 was characterized by the highest concentration of selected phytosterols, namely, campesterol and Δ5-avenasterol among the studied SCO_2_E oils. This may suggest the presence of these compounds in WSO in molecular combinations of a more polar nature (i.e., phytosteryl glycosides), which could shift the maximum of their extraction to a later period of time [27]. It was also observed that the carotenoid content decreased with the time of conducting the extraction and, generally, these compounds were less recovered with SCO_2_E than with CP. These results contradict those reported for raspberry seed oil recovered by both methods, where almost ten times higher concentrations of these compounds were found in SCO_2_E oil than with CP [5]. It should be noted, however, that in raspberry oil, carotenoids had a greater contribution to the total oil mass, which could have affected the proportions of extract components during processing. The low yield of squalene in SCO_2_E oils indicates that the applied processing parameters were not optimal for this component. This is confirmed by studies of amaranth oil particularly rich in squalene conducted by Wejnerowska et al. [28], which showed that the highest concentrations of this compound in SCO_2_E oil occur when applying a much higher temperature (up to 120 °C), but lower pressure and CO_2_ flow compared to this study [28]. Nevertheless, the cited research confirms the highest squalene yield in a short extraction time, similar to our results. In reference to a botanically similar oily raw material, *F. ananassa* seeds, WSO oil turns out to be a similarly rich source of phytosterols as CP-derived strawberry seed oil, and—irrespective of the production method used—is significantly richer in tocopherols. In the case of squalene, its presence in strawberry seed oil was undetectable in the research published so far [29].

The high content of linoleic acid and α-linolenic in the fatty acid composition acid may contribute to the high susceptibility of WSOs to oxidation. The prone to oxidation-prone bisallylic methylene position in linoleic and α-linolenic acid requires the lowest activation energy for induction of oxidation among fatty acids found in WSOs. In particular, this applies to α-linolenic acid, which has been shown to oxidize twice as fast as linoleic acid with the formation of highly unstable hydroperoxides [19]. In our study, we found no direct correlation between PUFA and α-linolenic acid contents and a decrease in the oxidative stability of oils in DSC tests. Among the oils obtained using the SCO_2_E method, SCO_2_30, which contained the highest amount of α-linolenic acid, was the most stable under isothermal conditions. The second most α-linolenic acid-rich oil was SCO_2_45, although its stability was much lower. These differences are attributable to the much higher content of tocopherols and phytosterols and the high antioxidant activity of SCO_2_30 oil compared to SCO_2_45. It is known that the addition of a sufficient level of antioxidants may decrease the kinetic rate of the induction period, resulting in higher values of OIT [30]. At the highest applied temperature of 120 °C under isothermal conditions and in non-isothermal tests, SCO_2_30 showed a stability similar to that of CP oil, which could be a result of its lower α-linolenic acid content and the simultaneous lower content of antioxidant substances. In accordance with our previous findings, it can be stated that the appropriate phytosterol-to-α-linolenic acid molar ratio of at least 1:100 better explains the resistance of SCO_2_30 to oxidation [31]. This association can be confirmed by the lower stability of SCO_2_15 when compared to SCO_2_30, despite its lower α-linolenic percentage and the highest content of tocopherols and polyphenols among the oils tested. In addition, this oil showed lower antiradical activity of the hydrophilic fraction of the oil than SCO_2_30, and, as previously shown, the contribution of polyphenols to the inhibition of oxidation induction is significant [31,32]. All studied oils recovered by SCO_2_E showed a long oxidation propagation period generally similar to that observed for CP oil, despite a higher α-linolenic acid content than in CP. This may be explained by the effect of antioxidants, especially polyphenols, which were at higher concentration in SCO_2_E oils. From this it can be inferred that the antioxidants were still able to scavenge the reactive free radical products under the oxidation conditions used and therefore they might prevent acceleration of the propagation rate [32].

Further investigations were focused on the safety aspects of using WSO in humans. The cytotoxicity of the tested WSO emulsions was assessed using the three assays most commonly used to detect cell viability or proliferative ability after exposure to the oil [33]. In the case of the SRB and LDH assays, which measure cell protein content and cell lysis, respectively, a cytotoxic effect of the tested oils was not demonstrated. This indicated that the composition of the oils at the concentrations used did not negatively affect the proliferative capacity of the cells, as further confirmed by the absence of signs of cell disintegration in the presence of WSOs. It is also worth mentioning that for connective tissue cells, such as fibroblasts, an increase in the amount of protein in the cell culture may also indicate transcriptional and translational activity leading to the synthesis of extracellular proteins such as collagen. Therefore, the amount of protein in the SRB assay does not necessarily translate into cell mass. In any case, in both SRB and LDH tests discussed, it was confirmed that the vital functions of fibroblasts were not inhibited and there were no necrotic effects. Similarly, the lack of an effect on fibroblast viability/proliferation was reported previously for raspberry seed oil obtained by the SCO_2_E method, which contained significantly more phytosterols and carotenoids than the WSOs currently under study [5]. A positive effect on fibroblast viability can be demonstrated by phytosterols at lower concentrations than applied emulsions with WSOs [34]. In our study on the bioactivity of *R. rugosa* and *R. canina* seed oils, we have also found a favorable effect on fibroblast viability, as revealed by the amount of protein in cells. All the mentioned oils from seeds of plants of the *Rosaceae* family are abundant in α-linolenic acid, as are WSOs. It can be assumed that the content of PUFA in WSOs, including α-linolenic acid, is a beneficial factor influencing membrane integrity and promoting cell viability, as demonstrated elsewhere for flaxseed oil emulsions and individual PUFA [34,35]. However, in the case of the MTT assay, a lower reductive effect on MTT was noted for WSOs collected after 15 and 30 min of SCO_2_E, which indicated their cytotoxic activity against NDHF. Both SCO_2_15 and SCO_2_30 oils contained considerable amounts of δ-tocopherol, for which higher accumulation was previously demonstrated in normal mouse mammary epithelial cells. [36]. Compared to α-tocopherol, this isoform undergoes less methylation, consequently retaining high lipophilicity in the cell, which was also associated with an increased effect on cellular DNA fragmentation. This is also confirmed by the morphological changes in the NDHF cells cultured with SCO_2_15 and SCO_2_30 oils. Interestingly, the IC_50_ for δ-tocopherol in MMT tests in a mouse mammary epithelial cell model was similar to that in the NDHF culture treated with WSO emulsion in the highest concentration (5%). However, the interpretation of MTT test results should consider the effect of the direct reductive potential of various antioxidants on tetrazolium salt without affecting mitochondrial succinate dehydrogenase activity, since it is well known that the results of the MTT assay can be inflated when various antioxidants are used [37]. In our study on the bioactivity of raspberry seed oil produced by the SCO_2_E, despite a similar δ-tocopherol content as in SCO_2_30 and SCO_2_45, no negative effect on NDHF cells was observed in the MTT assay [5]. It should be stressed, however, that in the mentioned study, we applied linear regression to estimate the cytotoxic effect, while the present study uses the Hill formula, which is a four-parameter logistic model suitable for binary dose–response assessments [38]. Moreover, in comparison to WSOs, the studied raspberry seed oil contained significantly more total carotenoids and tocopherols, whose possible chemical interaction with assay reagent cannot be ignored.

## 5. Conclusions

In summary it has been shown for the first time that oil recovered from *F. vesca* possesses a remarkable combination of bioactive compounds and its composition my differ, depending on the oil recovery method. WSOs obtained by CP and SCO_2_E are valuable sources of bioactive components, with their composition differing in terms of the contribution of specific fatty acids and the amount of micro-components. In addition, the extraction time with supercritical CO_2_ makes it possible to achieve a differentiated profile of phytosterols, tocopherols and carotenoids in the oil, but also influences the level of oxidation products. Therefore, our results demonstrate that both methods can be useful in wild strawberry’s waste recovery and contribute to the circular economy through the possible reintegration of WSO into food or biomedical applications. Our findings also indicate that cytotoxicity tests are necessary when choosing and fine-tunning the conditions of oils’ recovery.

## Figures and Tables

**Figure 1 foods-12-01852-f001:**
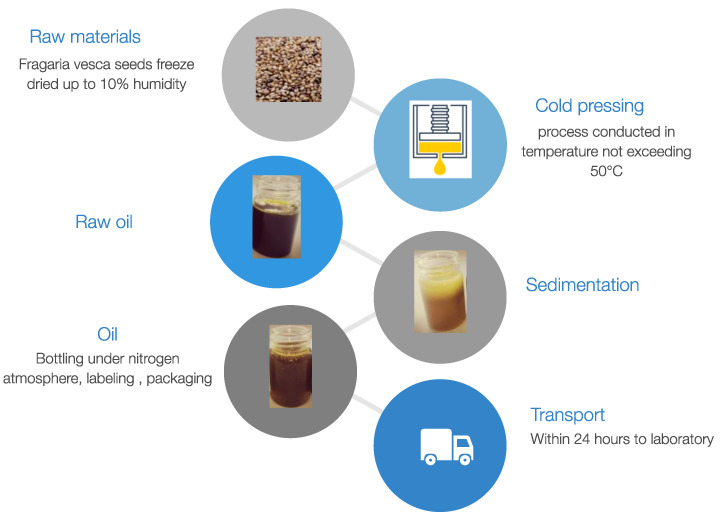
Cold-pressing process of *F. vesca* seeds.

**Figure 2 foods-12-01852-f002:**
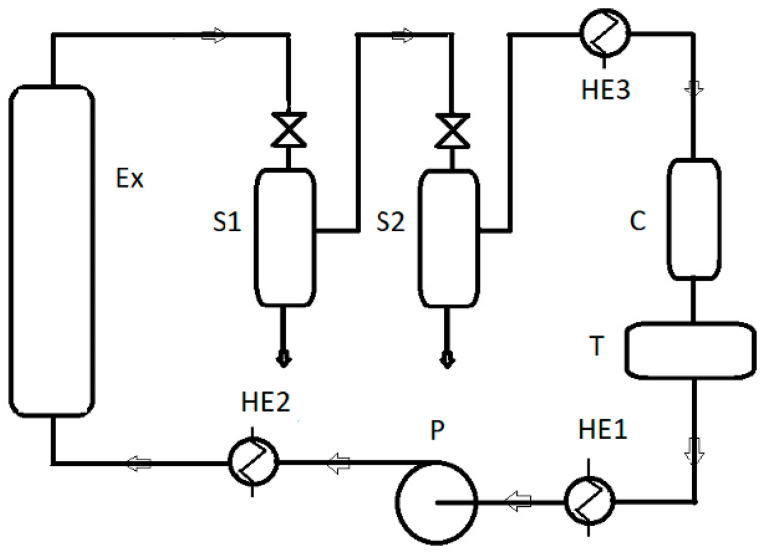
A pilot plant for supercritical CO_2_ extraction used in the experiment: Ex—extractor; S1, S2—separators; C—condenser; T—liquid CO_2_ tank; P—pump; HE1, HE2, HE3—heat exchangers.

**Figure 3 foods-12-01852-f003:**
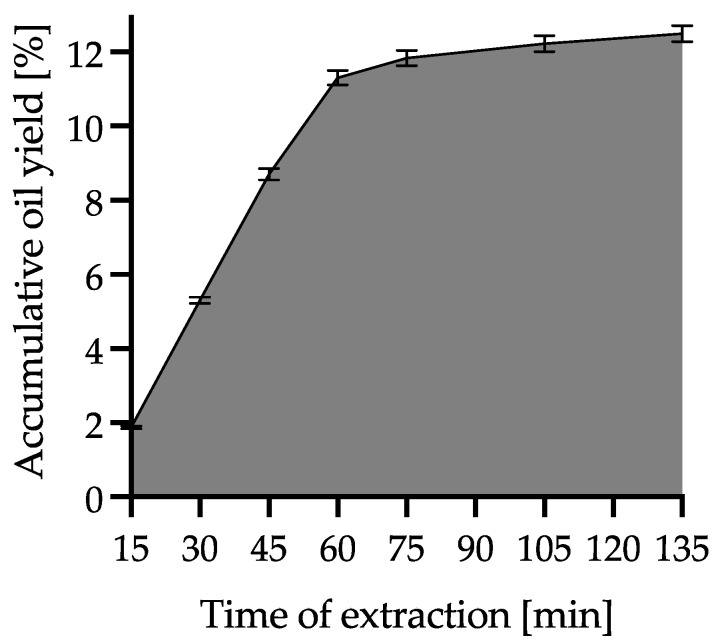
WSO accumulative yield obtained during SCO_2_E.

**Figure 4 foods-12-01852-f004:**
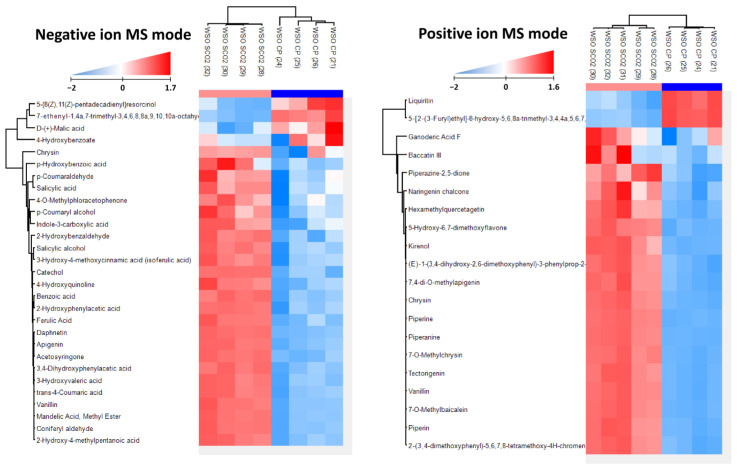
Hierarchical clustering analysis showing the differences in amounts of given compounds. Clustering was performed on both samples and compounds. Results show that samples clusters in proper groups, and compound clustering groups compound with similar profiles of polyphenol fraction of CP and SCO_2_15 WSOs. Similar analyses were performed on ESI/MS-negative and -positive mode data. In both cases, two main clusters of compounds are found with opposite concentration profiles.

**Figure 5 foods-12-01852-f005:**
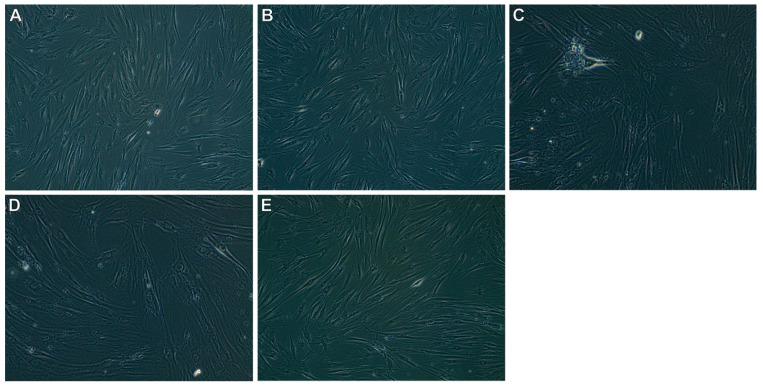
Cell morphology for 5% oils: (**A**) blank culture; (**B**) cold-pressed oil; (**C**) SCO_2_15; (**D**) SCO_2_30; (**E**) SCO_2_45.

**Table 1 foods-12-01852-t001:** Parameters of supercritical carbon dioxide extraction (SCO_2_E) process.

SCO_2_E Parameters	
Time of extraction [min]	135
Pressure of extraction [bar]	300
Temperature of extraction [°C]	40
Batch weight [g]	9850
CO_2_ pump flow [kg/h]	130

**Table 2 foods-12-01852-t002:** Fatty acid composition, antiradical capacity and quality parameters of wild strawberry seed oils (WSOs) (average ± SD) *.

	WSO							
Basic Characteristics	CP	SCO_2_15	SCO_2_30	SCO_2_45	SCO_2_60	SCO_2_75	SCO_2_105	SCO_2_135
Fatty acids [%]								
C18:0	2.756 ± 0.016 ^a^	2.040 ± 0.051 ^b^	1.91 ± 0.004 ^c^	1.972 ± 0.118 ^b^	2.606 ± 0.022 ^d^	2.746 ± 0.041 ^a^	2.622 ± 0.012 ^d^	2.571 ± 0.045 ^d^
C18:1 n-9	17.517 ± 0.014 ^a^	14.502 ± 0.422 ^bd^	12.714 ± 0.017 ^c^	12.784 ± 0.167 ^c^	14.125 ± 0.105 ^b^	14. 783 ± 0.202 ^d^	14.386 ± 0.199 ^bd^	14.143 ± 0.082 ^b^
C18:2 n-6	47.769 ± 0.025 ^a^	41.060 ± 0.145 ^b^	40.973 ± 0.73 ^b^	40.776 ± 0.138 ^c^	40.195 ± 0.023 ^de^	40.066 ± 0.023 ^d^	40.205 ± 0.075 ^de^	40.327 ± 0.061 ^e^
C18:3 n-3	30.840 ± 0.108 ^a^	41.296 ± 0.390 ^b^	43.304 ± 0.105 ^c^	43.124 ± 0.152 ^c^	41.218 ± 0.112 ^b^	40.04 ± 0.23 ^d^	40.748 ± 0.158 ^e^	41.080 ± 0.121 ^be^
PUFA	78.615 ± 0.194 ^a^	82.358 ± 0.527 ^b^	84.277 ± 0.172 ^c^	84.151 ± 0.278 ^c^	81.414 ± 0.162 ^d^	80.024± 0.298 ^e^	80.954 ± 0.17 ^d^	81.325± 0.084 ^d^
n-6/n-3	1.548 ± 0.007 ^a^	0.994 ± 0.006 ^be^	0.946 ± 0.001 ^c^	0.946 ± 0.001 ^c^	0.975 ± 0.003 ^d^	1.003 ± 0.006 ^b^	0.987 ± 0.002 ^e^	0.984 ± 0.001 ^de^
Antiradical capacity [mM TEAC/kg]								
Oil	1.031 ± 0.059 ^a^	1.243 ± 0.036 ^b^	1.255 ± 0.041 ^b^	0.919 ± 0.024 ^c^	0.767 ± 0.013 ^d^	1.210 ±0.029 ^b^	1.247 ± 0.020 ^b^	1.227 ± 0.010 ^b^
LP	0.775 ± 0.075 ^ad^	1.007 ± 0.060 ^b^	1.276 ± 0.050 ^c^	0.960 ± 0.031 ^b^	0.620 ± 0.027 ^a^	0.918 ± 0.150 ^bd^	0.891 ± 0.094 ^bd^	0.861 ± 0.085 ^bd^
HP	0.369 ± 0.030 ^a^	0.442 ± 0.027 ^be^	0.563 ± 0.027 ^c^	0.195 ± 0.008 ^d^	0.251 ± 0.054 ^d^	0.476 ± 0.025 ^b^	0.420 ± 0.031 ^abe^	0.404 ± 0.056 ^ae^
Quality parameters								
Acid Value (AV)	1.010 ± 0.014 ^ab^	1.288 ± 0.018 ^a^	0.838 ± 0.004 ^ab^	0.682 ± 0.004 ^b^	2.227 ± 0.003 ^c^	6.485 ± 0.238 ^d^	8.846 ± 0.445 ^e^	9.404 ± 0.506 ^f^
[mg KOH/g]								
Peoxide Value (PV)	1.678 ± 0.027 ^a^	1.322 ± 0.027 ^a^	2.960 ± 0.089 ^b^	1.352 ± 0.004 ^a^	8.768 ± 0.001 ^c^	7.963 ± 0.573 ^d^	12.910 ± 0.368 ^e^	11.655 ± 0.493 ^f^
[mEq O2/Kg]								

* Mean and standard deviations (SD) values were obtained from analyses of three samples of each fraction collected at particular time points. The statistical analysis was performed using the one-way ANOVA test, differences between oils were considered statistically significant at *p* < 0.05. The values in the same row that share the same superscript letter are not significantly different. TEAC—Trolox equivalent antiradical capacity.

**Table 3 foods-12-01852-t003:** Composition of minor bioactive compounds of cold-pressed and extracted supercritical CO_2_ wild strawberry oils (WSO) (mean * ± SD).

		WSO		
Minor Compounds [mg/kg]	CP	SCO_2_15	SCO_2_30	SCO_2_45
Phytosterols				
Campesterol	117.38 ± 24.01 ^ac^	86.50 ± 14.21 ^a^	175.29 ± 11.72 ^c^	206.64 ± 72.47 ^b^
Stigmasterol	25.46 ± 3.55 ^a^	20.24 ± 2.53 ^b^	ND ^c^	18.20 ± 1.01 ^b^
β-Sitosterol	3511.26± 179.39 ^a^	3382.36 ± 143.39 ^a^	5106.712 ± 478.57 ^b^	4252.24 ± 56.20 ^c^
Δ5-Avenasterol	133.2 ± 15.97 ^a^	86.40 ± 8.21 ^b^	114.00 ± 9.78 ^ba^	135.36 ± 5.48 ^ba^
Total	3787.31 ± 214.57 ^a^	3575.50 ± 143.56 ^a^	5396.100 ± 457.08 ^b^	4574.74 ± 177.50 ^c^
Squalene	31.4 ± 3.33 ^a^	23.13 ± 4.60 ^b^	11.12 ± 0.36 ^c^	12.71 ± 2.10 ^c^
Tocopherols				
α-Tocopherol	451.99 ± 179.57 ^a^	174.78 ± 7.26 ^b^	168.45 ± 33.55 ^b^	34.45 ± 4.46 ^b^
γ-Tocopherol	466.68 ± 169.7 ^a^	859.48 ± 107.07 ^b^	985.25 ± 210.36 ^b^	205.21 ± 43.03 ^a^
δ-Tocopherol	299.78 ± 60.79 ^a^	792.74 ± 68.69 ^b^	771.23 ± 93.96 ^b^	142.81 ± 26.43 ^a^
β-Tocopherol	ND	73.97 ± 12.56 ^b^	77.28 ± 19.82 ^b^	19.33 ± 2.09 ^c^
Total	1218.45 ± 408.14 ^abc^	1900.98 ± 79.10 ^c^	1795.08 ± 357.69 ^c^	401.80 ± 76.01 ^ab^
Carotenoids				
all-trans-β-Carotene	55.15 ± 12.67 ^a^	53.20 ± 14.78 ^a^	27.24 ± 7.55 ^b^	24.17 ± 2.04 ^b^
all-trans-Lutein	14.4 ± 2.73 ^a^	10.13 ± 1.53 ^b^	5.413 ± 1.41 ^c^	3.23 ± 0.29 ^c^
all-trans-Zeaxanthin	39.88 ± 0.37 ^a^	39.52 ± 0.14 ^a^	39.43 ± 0.23 ^b^	39.07 ± 0.01 ^b^
all-trans-α-Cryptoxanthin	14.36 ± 0.07	ND	ND ^c^	ND
Total	123.78 ± 15.09 ^a^	102.85 ± 16.18 ^a^	77.96 ± 9.19 ^b^	66.47 ± 2.34 ^b^
Total Polyphenols	15.12 ± 1.40 ^a^	58.45 ± 2.58 ^a^	40.48 ± 1.93 ^b^	45.32 ±0.06 ^b^

* Mean and standard deviations (SD) values were obtained from measurements of three analytical replicates for each fraction combined at particular time points. The statistical analysis was performed using the one-way ANOVA test, differences between oils were considered statistically significant at *p* < 0.05. The values in the same row that share the same superscript letter are not significantly different. ND—not detected.

**Table 4 foods-12-01852-t004:** DSC results of oxidative stability for selected WSOs obtained in isothermal and non-isothermal conditions.

Temperature Applied	Oxidation Parameter		WSO		
		CP	SCO_2_15	SCO_2_30	SCO_2_45
Isothermal conditions, time [min]					
100 °C	OIT	386.07 ± 29.34 ^a^	225.66 ± 23.03 ^b^	443.92 ± 44.95 ^c^	6.85 ± 0.44 ^d^
	Propagation	96.78 ± 5.58 ^a^	86.97 ± 5.18 ^b^	72.01 ± 5.07 ^c^	58.89 ± 3.11 ^c^
110 °C	OIT	134.21± 4.36 ^a^	128.66 ± 9.02 ^b^	184.76 ± 7.11 ^c^	6.86 ± 0.44 ^d^
	Propagation	59.68 ± 3.31 ^a^	55.67 ± 15.43 ^b^	70.16 ± 9.96 ^c^	58.90 ± 3.10 ^a^
120 °C	OIT	84.23 ± 2.83 ^a^	61.12 ± 3.61 ^b^	84.05 ± 7.40 ^a^	0.09 ± 0.01 ^c^
	Propagation	15.17 ± 3.56 ^a^	12.07 ± 0.97 ^a^	22.94 ± 2.05 ^b^	19.90 ± 2.18 ^b^
Ea [kJ/mol]		100.41 ± 5.85 ^a^	85.23 ± 12.70 ^b^	105.00 ± 5.91 ^a^	ND
Non-isothermal conditions, temperature [°C]					
Rate 5 °C/min	T_on_	164.42 ± 1.32 ^a^	157.57 ± 3.96 ^b^	167.33 ± 0.68 ^a^	144.28 ± 1.54 ^c^
	Propagation	157.69 ± 2.58 ^ab^	161.53 ± 3.26 ^a^	153.21 ± 1.01 ^b^	177.84 ± 1.99 ^c^
Rate 10 °C/min	T_on_	174.03 ± 0.58 ^a^	167.92 ± 3.19 ^a^	175.46 ± 5.08 ^a^	145.86 ± 8.97 ^b^
	Propagation	157.15 ± 1.46 ^ab^	159.95 ± 3.42 ^ab^	151.79 ± 1.16 ^a^	170.50 ± 15.11 ^b^
Rate 15 °C/min	T_on_	184.58 ± 1.77 ^a^	181.99 ± 3.81 ^a^	185.85 ± 2.01 ^a^	164.56 ± 5.72 ^b^
	Propagation	153.43 ± 2.11 ^a^	88.04 ± 7.26 ^b^	151.32 ± 2.43 ^a^	169.40 ± 2.70 ^c^

OIT—Oxidation induction time; Ea—Activation energy; T_on_—Oxidation induction temperature; Propagation—the difference between oxidation induction (OIT or Ton) and the termination of oxidation. ND—not determined. The values in the same row that share the same superscript letter are not significantly different.

**Table 5 foods-12-01852-t005:** Cytotoxic activity of the tested WSOs against the normal cell line—NHDF determined in the MTT test and expressed as the cytotoxic concentration (CC_50_ ± SD) which causes a 50% reduction of viable cells. SD is the standard deviation.

WSO	MTT Assay	LDH Assay	SRB Assay
	CC_50_ ± SD (%)		
CP	Non-toxic	Non-toxic	Non-toxic
SCO_2_15	3.35 ± 0.67%	Non-toxic	Non-toxic
SCO_2_30	5.49 ± 0.91%	Non-toxic	Non-toxic
SCO_2_45	Non-toxic	Non-toxic	Non-toxic

**Table 6 foods-12-01852-t006:** In vitro cytotoxicity cell morphology results after treatment with WSOs after 24 h. The evaluation was carried out according to ISO 10993-5. The rating represents the mean of three independent observations.

WSO	Description	Grade
CP (5%)	Discrete intracytoplasmic granules, no cell lysis	0
SCO_2_15 (5%)	Around 15% of the cells are round and loosely attached, with culture density lower than the blank sample	1
SCO_2_30 (5%)	Around 5–10% of the cells are round and loosely attached, with culture density lower than the blank sample	1
SCO_2_45 (5%)	Discrete intracytoplasmic granules, no cell lysis	0

## Data Availability

All related data and methods are presented in this paper. Additional inquiries should be addressed to the corresponding author.

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
