# Peer review of "Evaluation of Recovery Methods for Fragaria vesca L. Oil: Characteristics, Stability and Bioactive Potential"

_foods, 2023, doi:10.3390/foods12091852_

Round 1
Reviewer 1 Report
Comments and Suggestions for Authors
[1] Please add more information or figures on the experimental apparatus used in cold pressing (CP) and supercritical carbon dioxide extraction (SCO2E).
[2] Please justify the operating conditions contained in Table 1.
[3] Change Kg for kg.
[4] Figure 1 (The Profile of Polyphenolic Fraction) is challenging to interpret. Have a more friendly alternative figure?
Author Response
Review #1
We appreciate your time and effort in reviewing our article, and we are grateful for Your valuable comments. In line with your suggestions, we amended the manuscript as listed below.
[1] Please add more information or figures on the experimental apparatus used in cold pressing (CP) and supercritical carbon dioxide extraction (SCO2E).
The schematic descriptions of the experimental apparatus are now added into manuscript in Material & Methods section as listed below:
Figure 1 Cold-pressing process of F.vesca seeds
Supercritical carbon dioxide extraction pilot plant (SCO2E):
Fig. 2. Schematic of a pilot plant for supercritical CO2 extraction, Ex-extractor, S1, S2-separators, C- condenser, T - liquid CO2 tank, P- pump, HE1, HE2, HE3 – heat exchangers.
[2] Please justify the operating conditions contained in Table 1.
Due to the planned cooperation with the industry in the production of wild strawberry seed extract, cost optimization was carried out, which showed that the optimal parameters were pressure - 300 bar, temperature - 40 0C, CO2 consumption - 30 kg/kg, which resulted in a time of 135 minutes. The graph (Fig. 3) shows that the extraction could be completed even after 105 minutes. The practical criterion for the completion of the process was a cumulative increase in the mass of the extract below 0.3%. Missing information were added to the manuscript in Material & Methods section.
Figure 3 Wild strawberry seed oil SCO2E curve.
[3] Change Kg for kg.
Thank you for your notice. We changed Kg to kg.
[4] Figure 1 (The Profile of Polyphenolic Fraction) is challenging to interpret. Have a more friendly alternative figure?
Thank you for your comment. The improved figure was now incorporated into manuscript.

Reviewer 2 Report
Comments and Suggestions for Authors
The manuscript focuses on the evaluation of Fragaria vesca L. oil recovery methods and their characteristics, stability and bioactive potential. The methodology seems to have been carried out adequately, but it is not clear how many extractions were performed for the analysis of the experiments. The results for minor compounds and oil stability of WSO-SCO2 15 and 30 are interesting, but I'm not sure there were enough extraction numbers. I suggest that authors send complementary data of all extracts to the editor.
Also, some errors are described below:
• There are minor spells that should be checked in the text, especially like CO2.
• In section 3.1.1. fatty acids should be identified according to Table 2 (C18:0, 1, 2 or 3) to be clearer for readers.
Author Response
Review # 2
We appreciate all the comments listed by the Reviewer that contributed to the improvement of manuscript.
- The manuscript focuses on the evaluation ofFragaria vesca oil recovery methods and their characteristics, stability and bioactive potential. The methodology seems to have been carried out adequately, but it is not clear how many extractions were performed for the analysis of the experiments. The results for minor compounds and oil stability of WSO-SCO2 15 and 30 are interesting, but I'm not sure there were enough extraction numbers. I suggest that authors send complementary data of all extracts to the editor.
Prior to the described experiment, several commercial extractions of strawberry/wild strawberry seeds were carried out at the request of a business partner, the purpose of which was to produce an oil extract and its evaluation. These studies allowed to select the extraction parameters. At the same time, during the sampling, a change in the color of the extract was observed depending on the time of sampling. This was an inspiration to conduct detailed analytical studies of individual samples taken at specific moments in time. In the considered case, due to the large charge (9850 g) and extraction costs, one full experiment was performed, taking 3 samples for analysis from each sample taken at a certain time. Individual samples were carefully mixed and prepared for analysis in accordance with the methodology used. It should be noted that the samples taken at certain times during the first hour of extraction had a weight of about 300 - 400 g each. A total of 1229.38 g of extract was produced. All the missing information are now written down in Material & Methods section of the manuscript.
Also, some errors are described below:
- There are minor spells that should be checked in the text, especially like CO2.
Thank you for you notice. All spells error were corrected in the text.
- In section 3.1.1. fatty acids should be identified according to Table 2 (C18:0, 1, 2 or 3) to be clearer for readers.
Thank you for your comment. Fatty acids are now identified according to the Table 2.
Reviewer 3 Report
Comments and Suggestions for Authors
This manuscript describes an in-depth investigation providing new data on the composition and quality of wild strawberry seed oil (WSO) recovered by two methods (CP and SCO2E). In addition, cell toxicity tests were carried out, assessing WSO impact on cell membrane integrity, cell mitochondria metabolic activity and overall cell viability/proliferative ability using human dermal fibroblast cells.
The paper answers some questions and advances knowledge in the field, given that literature-derived data on the composition of WSO do not include information on the content of bioactive oil’s micronutrients, such as phytosterols, tocochromanols, carotenoids, and phenolic compounds.
This research provides new information on the phytochemical and oxidative stability characteristics of WSO recovered by CP and SCO2E.
The introduction provides an overview of the current research on this topic and the motivation for this research. Given that the seeds are part of the residual biological mass resulting from the processing of wild strawberry fruits, please add a sentence highlighting the contribution of this research to the circular economy through the possible reintegration of WSO into food or biomedical applications.
The study is well designed and well written, the methodology is provided in sufficient detail. The results are well illustrated and well aligned with the purpose of this paper. The results presented in Table 2-5 and Figure 1 and 2 prove every detail discussed in the paper. Please, correct the number of figure 3 (it is actually figure 2).
Please make the conclusions more consistent, by highlighting the innovative aspects of this study that support its added value.
The references cited are relevant to the research topic.

Author Response
Review #3
The manuscript has been revised and undergone changes to meet the Reviewer comments. We would like to thank you for your review that helped us to improve the manuscript
This manuscript describes an in-depth investigation providing new data on the composition and quality of wild strawberry seed oil (WSO) recovered by two methods (CP and SCO2E). In addition, cell toxicity tests were carried out, assessing WSO impact on cell membrane integrity, cell mitochondria metabolic activity and overall cell viability/proliferative ability using human dermal fibroblast cells.
The paper answers some questions and advances knowledge in the field, given that literature-derived data on the composition of WSO do not include information on the content of bioactive oil’s micronutrients, such as phytosterols, tocochromanols, carotenoids, and phenolic compounds.
This research provides new information on the phytochemical and oxidative stability characteristics of WSO recovered by CP and SCO2E.
- The introduction provides an overview of the current research on this topic and the motivation for this research. Given that the seeds are part of the residual biological mass resulting from the processing of wild strawberry fruits, please add a sentence highlighting the contribution of this research to the circular economy through the possible reintegration of WSO into food or biomedical applications.
Thank you for you notice we added a following sentence into manuscript:
Introduction:
“Washing and processing fruits generates leftovers liquid contains a range of biomass that can be recovered for biomedical applications.”
Conclusions:
“Therefore, our results demonstrate that SCO2E can be a useful method in wild strawberry’s waste recovery and contribute to the circular economy through the possible reintegration of WSO into food or biomedical applications.”
- The study is well designed and well written, the methodology is provided in sufficient detail. The results are well illustrated and well aligned with the purpose of this paper. The results presented in Table 2-5 and Figure 1 and 2 prove every detail discussed in the paper. Please, correct the number of figure 3 (it is actually figure 2).
Thank you for your comment the correct numbers of figures are now incorporated into manuscript.
- Please make the conclusions more consistent, by highlighting the innovative aspects of this study that support its added value.
Thank you for your comment, the conclusions have been changed from:
“This work offers new information about phytochemical and oxidative stability characteristics of oil recovered form F.vesca seeds by two methods that are considered to be the least degrading of the bioactive components of vegetable oils: CP and SCO2E. WSOs obtained by both methods are valuable sources of bioactive components, with their composition differing in terms of the contribution of specific fatty acids and the amount of micro-components. In addition, the extraction time with supercritical CO2 makes it possible to achieve a differentiated profile of phytosterols, tocopherols and carotenoids in the oil, but also influences the level of oxidation products. Our results demonstrate that SCO2E can be a useful method in targeted valorization of post-production residues of wild strawberry processing. They also show that cytotoxicity tests are necessary when choosing and fine-tunning the conditions of oils’ recovery. The method of recovery of oil from seeds obtained as a waste product of the food industry must be considered when producing oils intended for improving health and of good quality.”
to
“In summary it has been shown for the first time that oil recovered from F.vesca possesses a remarkable combination of bioactive compounds and its composition my differ depending on the oil recovery method. WSOs obtained by CP and SCO2E are valuable sources of bioactive components, with their composition differing in terms of the contribution of specific fatty acids and the amount of micro-components. In addition, the extraction time with supercritical CO2 makes it possible to achieve a differentiated profile of phytosterols, tocopherols and carotenoids in the oil, but also influences the level of oxidation products. Therefore, our results demonstrate that both methods can be a useful in wild strawberry’s waste recovery and contribute to the circular economy through the possible reintegration of WSO into food or biomedical applications. Our findings also indicate that cytotoxicity tests are necessary when choosing and fine-tunning the conditions of oils’ recovery.”
The references cited are relevant to the research topic